# On the Streets of Paris: The Experience of Displaced Migrants and Refugees

## Madeleine Byrne

Independent Researcher, Paris, France; madeleine.byrne@eduservices.org

**Abstract:** In the wake of the demolition of the "The Jungle" at Calais, northern France, in October 2016, more than three thousand asylum seekers, refugees and other informal immigrants at any given time live in informal tent cities throughout the city's northern areas. These makeshift camps appear to manifest a central issue in the French asylum system, that is applicants after making a claim for protection, and awaiting a hearing or decision, receive next to no formal support (financial, or residential) and are largely left to fend for themselves. Not all of the camp residents are asylum seekers wanting to stay in France. Some are migrants (or asylum seekers) en route to the United Kingdom; others are refugees who received French protection, with no housing. Between 2015–2017 there were multiple outbreaks of scabies in these tent cities leading to sanitation workers refusing to work in their vicinity. The current Covid-19 crisis has, moreover, further exacerbated concerns about the health of the unhoused asylum seekers and migrants in Paris - unaccompanied minors, in particular. This article will consider the repeated displacement, or dispersal, of this population in terms of the changing "politics of immigration"and policing in France under President Emmanuel Macron. In order to present the broader social context, it will also describe current events in Paris, including Macron government's legislation relating to asylum/immigration, policing and more, amid the Covid-19 health crisis.

**Keywords:** refugees; immigration; asylum policy; Paris; France

On the 22 November 2020, French police destroyed a migrant and refugee camp at the Place de la République in central Paris. The *Police Nationale*, whose remit covers large urban areas, and the *Gendarmerie Nationale*—an agency with a military rather than civil function, presided over by the Ministry of the Interior—were joined by the *Brigades anti-criminalité* (BAC), a police force based in the mostly low-income, migrant *banlieue* neighbourhoods surrounding French cities.

Within 45 min of the migrants and refugees' arrival at Place de la République, the police applied pressure on the group—upturning the flimsy tents, roughly turning those inside them onto the pavement and, in some instances, beating people with truncheons who were already lying prone on the ground—then forced them to leave. Journalist Rémy Buisine, from the *Brut* news service, reported that the group of mostly Afghan men left by 10.30/11 pm after multiple police assaults and the confiscation of their tents.[1] Buisine also noted that the men had previously been moved from the Porte Saint Denis, in the neighbouring 10th *arrondissement*, after police tear gassed them.

"For the last week, (the migrants and asylum seekers) have been sleeping under bridges," Reza Jafari—president of the NGO, Children from Afghanistan—told Buisine in French "(For the police) it's like they are hunting people. Each time they arrive, they beat people. They humiliate them." Buisine asked where Jafari and the others would sleep that night. "We're going to try and go towards the *Mairie de Paris* (Paris Town Hall) knock on the door of the *Mairie de Paris.*"

---

1    Rémy Buisine, "Comment l'installation et l'évacuation d'un camp de réfugiés à Paris a dégénéré" (tweet) 24 November 2020. https://twitter.com/ brutofficiel/status/1331305012731973639. (Accessed 24 November 2020) All translations from the French are author's own.

Place de la République is the preferred location for organised political protests in Paris and has a particularly strong association with activists of the Left. It was the site of nightly protests in 2016 as part of the *Nuit debout* movement, and where millions of Parisians gathered following the 13 November 2015 terrorist attacks. Thus, the selection of the site by the migrants and their supporters was intentional and political. Media coverage of the event did not explicitly engage with the motivation of the men—or their supporters—or ask them to explain why they set up their tents in this public place, perhaps because they automatically assumed the action to be activist in intent. To understand this, you need to remember where these men usually set up their camps—in squalid ditches; under aerial railway lines; beside highways, in the northern parts of the city where few people shopping at Place de la République would choose to venture.

Outside the fact that it is a large open space—a rarity in over-crowded Paris—Place de la République is a preferred location for protests and demonstrations because it is at the intersection of three arrondissements—the 3rd, 10th and 11th—and therefore close to multiple transport options. Out of these three districts, only the 10th might be called impoverished and a more "expected" or predictable locale for non-French people to sleep rough on the street in large groups. Around the square, there are chain stores for sporting equipment, a mid-range hotel, and large restaurants: it is not a particularly attractive place for tourists, it is a place where Parisians go shopping, or meet their friends.

Significantly, as reported by *Le Parisien,* this sense of the migrants and refugees upsetting what might be called the natural order of things was made explicit by the police. The journalist Rémy Buisine later reported that the group ended up at the Porte d'Aubervilliers area of northern Paris, on the far edge of the city, where, after 1 am, they were served meals by French volunteers and left to sleep on the street, many without covering or protection. "The police told them: We will leave you alone, if you don't come back to Paris," Mael de Marcellus, the coordinator of the refugee rights group, Utopia 56's Paris branch, was quoted in the newspaper. Utopia 56 is a national organisation that works with refugees and asylum seekers in Calais, Lille, Paris, Rennes, Toulouse and Tours and was founded in 2016 to work with people stranded at The Jungle in Calais.

Marcellus then spelled out what might have been the motivation of the activists in organising the event: an attempt to force Parisians to acknowledge the existence of thousands of people who are usually unseen and ignored. Referring to the police effort to force the men out of central Paris to the northern *banlieue,* he said: "We want to make the asylum seekers/migrants[2] invisible, get them to leave the nice neighbourhoods (*beaux quartiers*) to send them to the working-class areas . . . It's as dehumanising as possible for those who are already victims. (Anthony Lieures 2020)"[3] Buisine, in his report, also interviewed Yann Manzi, co-founder of Utopia 56 who said: "It's violent like this every night for these people who are rendered invisible," he said. "Today we wanted to show you what these people live every night, hidden under bridges, hounded by our police, treated as if they were animals, as if they were terrorists .... It's a disgrace (*honte*); I no longer recognise my own country."

Rather than being a rare occurrence, migrants and asylum seekers settling in public areas, and then being forced out by the police, is commonplace in Paris. According to a November 2019 *Guardian* report, there have been "more than 30 clearances (of such camps) in the capital over the last four years, but each time hundreds more people end up back on the streets shortly after (Chrisafis 2019)." The article referred to "3000 people sleeping on pavements and under bridges and canals in northern Paris and Seine-Saint-Denis,

---

[2] Marcellus uses the word "exilé" to refer to the people who were expelled from the square. Médecins Sans Frontières also uses the terms in reports on the subject.

[3] Mael de Marcellus quoted in "Evacuation de migrants à Paris: «On se sert de la banlieue pour repousser les problèmes»" (Anthony Lieures 2020).

with scant access to running water and no showers" in locations with appalling sanitary conditions, infested by rats (Foulon 2019)[4].

Sans Frontières (2019) noted in November, meanwhile, that over the previous four years, there had been 59 evacuations of such informal camps in the city's north-east, many of which were accompanied by police violence. (The report continued: "59 times a plan for change was promised but never eventuated; 59 times, and in the weeks that followed, we watched as the men, women and children returned to the street, forced to separate and isolate themselves to avoid police harassment".) Yet another article claimed that 65 or 66 camps in Saint-Denis—the *banlieue* to the north of Paris, which is located in the region with the highest level of poverty in France, Seine-Saint-Denis (C.G. 2020)—had been dismantled over a five-year period and now had a semi-permanent police positioned nearby.

Whether it is 30 such clearances in four years across the northern districts of the city, or the neighbouring *banlieue*, 59, 65 or 66 ultimately is not the issue. The Place de la République events are standard in Paris and have been for the past five years, if not longer. What distinguished them was the fact that the police were filmed using force against the asylum seekers and migrants at a time when the issue of police violence in France was a major news story and source of public concern. This might be why the conservative[5] Minister of the Interior Gérald Darmanin tweeted the following just before midnight the same day:[6]

> "Certain images from the expulsion of the unauthorised migrant camps at the Place de la République are shocking. I have asked for an official report (*rapport circonstancié*) of the relevant facts from the Police Prefect by tomorrow midday. I will make decisions on further action as soon as I receive it."[7]

By the following Wednesday, according to *The Guardian*, the Minister for the Interior had changed his position markedly:

> "These people were mostly without papers and illegally installed on Place de la République. This clearing [of the camp] was completely legitimate," he told France 2. "There were unacceptable actions from some police officers . . . these have to be looked into . . . those who messed up will be punished."

---

4   A contemporaneous report in *Le Parisien* included a map of the various camps throughout northern Paris and neighbouring Saint-Denis. The 3000 people, according to the report, were set up close to the exits for the ring road around the city: the biggest settlement was at Porte D'Aubervilliers with 512 tents and 15 huts—at Avenue du Président-Wilson, in Saint-Denis, there were 279 tents and seven huts; migrants and refugees had also set up camps at Porte de la Villette.

5   This word "conservative" is not entirely accurate here as Darmanin is central to the Macron government's campaign to—depending on your perspective—target or identify "Islam" and "extremist Islam" as a threat to "French Republican values". This dynamic is not new in France. In fact, Interior Ministers have taken on this role of targeting Islam, residents of mostly immigrant neighbourhoods, or Muslims in general since the 1970s as a way of shoring up political support and minimising the potential impact of far-right parties; see, for example, the then Interior Minister Nicolas Sarkozy, who later became President. During a 20 October TV show, for example, Darmanin said how he was "always shocked" to see food products associated with particular religious communities on supermarket shelves, to conclude that such a practice marks the start of the separation of communities (*communautarisme*). Darmanin did not explicitly refer to Muslims—even if the term *communautarisme* is not easily translated, "multiculturalism" is the closest meaning, although unlike the word "multiculturalism" it is rarely used outside a highly politicised context and is generally seen to be negative, threatening and usually used in relation to Muslim people in France —but the news report immediately afterwards showed footage of Halal and Kosher food products. The news report, for example, had the tags: health, food, Halal meat. Moreover, Darmanin made these comments four days after the brutal murder of a French middle-school teacher by an Islamist terrorist, Samuel Paty, on 16 October that prompted mass grief and enormous distress in France. (as cited in Peyrout et al. 2020).

6   Mathieu Hanotin, president of the Socialist Party for the territory Plaine Commune and mayor of Saint-Denis said in terms of the Place de la République clearance and the fact that the migrants and refugees were pushed out of central Paris: "I am revolted and scandalised. To use the *banlieue* to remove problems, is unacceptable. So that as soon as (the refugees and migrants) cross the border, we don't care for them anymore? When there is a problem, we resolve it, we don't use Seine-Saint-Denis to hide it . . . " Quoted in "Evacuation de migrants à Paris". Stéphane Troussel—Socialist Party President of Seine-Saint-Denis, was equally scathing: "What took place at Place de la République was a hunt of migrants organised by the Prefect of the Police and it's time to return to reason. The use of force was disproportionate, disgraceful—dignity and humanity are not options in our country." (as cited in Sans Frontières 2019).

7   Quoted in Buisine, Comment, 24 November 2020.

### 1. The "Global Security Law" (*Loi Relative à la Sécurité Globale*) and the Broader Socio-Political Context in Paris

From September 2020 onwards, the French President Emmanuel Macron—and members of his government—made several public pronouncements targeting "Islamist separatism" in France, even if what this meant exactly remained vague.[8] President Macron (2020)'s 2 October speech, entitled "Fight against separatism, the Republic in action" included the following explanation of the problem (this English translation is provided by the government):

"What we must tackle is Islamist separatism. A conscious, theorized, political-religious project is materializing through repeated deviations from the Republic's values, which is often reflected by the formation of a counter-society as shown by children being taken out of school, the development of separate community sporting and cultural activities serving as a pretext for teaching principles which aren't in accordance with the Republic's laws. It's indoctrination and, through this, the negation of our principles, gender equality and human dignity . . .

The problem is this ideology, which claims that its own laws are superior to the Republic's. And as I've often said, I'm not asking any of our citizens to believe or not believe or believe a little or moderately—that's none of the Republic's business. I'm asking every citizen, of all religions and none, to abide wholeheartedly by all the Republic's laws. And in this radical Islamism—since this is at the heart of the matter let's talk about it and name it—a proclaimed, publicized desire, a systematic way of organizing things to contravene the Republic's laws and create a parallel order, establish other values, develop another way of organizing society which is initially separatist, but whose ultimate goal is to take it over completely. And this is gradually resulting in the rejection of the freedom of expression, freedom of conscience and the right to blaspheme, and in us becoming insidiously radicalized. Nearly 170 people, to give just one example, are being monitored here, in [the French department of] Yvelines, for violent radicalization. Sometimes this goes as far as going and waging jihad. We know that 70 young people in this department left for Syria, and it's often children of the Republic who stray down this path, even going as far as actually taking action and trying to cause bloodshed or sometimes worse. It's also this path whose manifestations we saw again last Friday, near the premises of Charlie Hebdo."

In December, his government introduced a law "reinforcing Republican values" which included a series of measures intended to defeat, in the French Prime Minister, Jean Castex's words the "the pernicious ideology that goes by the name of Islamist radicalism" (RFI/Mike Woods 2020). The law introduced measures covering a wide range of matters: doctors would be fined or jailed for performing so-called "virginity tests"; people practicing polygamy, which is illegal in France, would be refused residency permits; mosques would be required to announce any foreign funding over €10,000 and register as places of worship.[9]

Detailed analysis of this law—and what it represents in terms of French society, past and present—is outside this article's scope, even if there are many contentious, even questionable, claims included in Macron's speech. Few people in France, outside those with far-right sympathies would argue that there is a unified push among "Islamist separatists"—or those with "radical Islamist" views—to either contravene the country's laws, or overthrow them in order to take over the reins of power. France's almost five million Muslims are

---

8 List of terrorist incidents in France, Wikipedia; https://en.wikipedia.org/wiki/List_of_terrorist_incidents_in_France. (accessed on 25 November 2020).

9 RFI/Mike Woods, "French government". See also Momtaz (2020), "5 things to know about France's bill to combat Islamist radicalism".

generally seen to be among the most "integrated" in Europe, with very few holding views antithetical to what might be called "Republican" values.[10]

Ongoing acts of terrorist violence, most often committed by young men born in the country, grounded by an allegiance to Daesh/Islamic State, or al-Qaeda is a serious issue in France. It is impossible to minimise the effect that such violence has on people living in the country in a psychological or social sense, both because of its extreme brutality—an 85-year-old priest murdered at the altar of his church in a small town in Normandy, 2016, or the dozens of children killed as a truck careered into people celebrating Bastille Day in Nice in an assault that killed 86 people and injured a further 458 people more the same year—or what appears to be its constant nature.

Whether or not this violence is a direct consequence of an organised push towards "Islamist separatism", as suggested by Macron's speech or whether a law such as the one his government is proposing is the most effective way to respond to this problem is a matter of debate with France's borders and elsewhere. Most of the perpetrators of such violence are isolated individuals, often French converts or young men with a history of petty crime, acting in rebellion against what they see to be the appeasing nature of organised religion in the country and not being indoctrinated by local mosques: their combat is transnational in nature, even if their victims are French.

The forced evacuation of the group of mostly Afghan nationals from the Place de la République occurred one month after a "Global Security Law" (*loi relative à la sécurité globale*) was tabled in the French Parliament (on 20 October 2020).

Two members of Emmanuel Macron's party *La République en marche* (LREM) sponsored the law: Alice Thourot, a lawyer and politician representing the eastern Rhône-Alpes region and Jean-Michel Fauvergue, a former police commissioner and director of the country's elite tactical unit, RAID between 2013 and 2017. RAID, which stands for "Recherche, Assistance, Intervention, Dissuasion" and was established in 1985 to counter all forms of criminality, such as the "organised crime of gangs, terrorism and hostage-taking."[11]

Fauvergue, was present during the Bataclan Theatre massacre in 2015, where a team of mostly French-born gunmen and suicide bombers killed 90 people, and injured hundreds more. He also directed the operation against Amedée Coulibaly the previous January, where, after taking 19 people hostage, Coulibaly killed four, one day after he had shot a police officer. In the 1990s, Fauvergue led anti-terrorist units in New Caledonia, French Guiana, and acted as the French Embassy's security attaché at Bamako, Mali and Libreville, Gabon. On his return to France, Fauvergue was responsible for projects targeting "people smuggling" and the employment of irregular migrants.[12]

The law, sponsored by Fauvergue and Alice Thourot, triggered mass protests in France, primarily for the clause that introduced a fine of €45,000 and possible year-long jail term for anyone who filmed a police officer, showing their face or identity "with the aim of damaging their physical or psychological integrity". The French government has since stated that they will rework this provision, but the desire to punish people filming police officers and then making it public jarred with the claim that "free speech" was a core Republican value, as repeatedly stated after the murder of the teacher Samuel Paty by an Islamist terrorist that occurred four days earlier.

---

[10]　There have been numerous reports and studies indicating that the vast majority of French Muslims hold attitudes that resemble the majority French population on subjects relating to social attitudes and secularism. For instance, a 2005 study—Sylvain Brouard et Vincent Tiberj, *Français comme les autres? Enquête sur les citoyens d'origine maghrébine, africaine et turque*, Préface de Pascal Perrineau, Les Presses de Sciences Po (collection Nouveaux débats), 2005—found that one-fifth of the respondents from West Africa, Turkey and North Africa had no religion, as compared to 28% of the French population. Moreover, 69% of those surveyed thought that democracy worked well in France as compared to 59% of the general population and most dramatically more than 80% believed that the word secularism was very, or quite positive and a similar percentage agreed with the sentence: "in France, secularism allows people of different faiths to live together." The study found that younger people in the survey tended to be more religious than others in the group, but that this religious observance was seen to be deeply personal (rather than political) and declined with age. The authors concluded that "rather than being a danger, the idea of *le communautarisme is a fantasy.*" (as cited in Lamchichi 2006).

[11]　Recherche, assistance, intervention, dissuasion (RAID), Wikipedia, https://fr.wikipedia.org/wiki/Recherche,_assistance,_intervention,_dissuasion. (accessed on 26 November 2020).

[12]　Jean-Michel Fauvergue, Wikipedia, https://fr.wikipedia.org/wiki/Jean-Michel_Fauvergue. (accessed on 26 November 2020)

Moreover, the fine's amount, which exceeds the French median salary—according to *l'Institut National de la Statistique et des Études Économiques* (INSÉÉ), the average salary in France in 2020 is €37,000 per year[13] even if many receive nothing close to that—reinforced the impression that the law was excessive and overly punitive, especially since those who experience police brutality are those most likely to be working in low-paid jobs.

What is immediately striking about the "Global Security Law", and also the public comments by President Macron and his government, is the way that it merges relatively anodyne issues (from my non-French perspective)—Muslim women wanting single-sex bathing times at a swimming pool, for example—with the illegal and extremely violent. The opening statement—or justification—for the "Global Security Law" as mentioned above maintains this tendency of blurring (apparently) disparate issues into an argument relating to counteracting "insecurity" in France. Note how minor offences are posited as comparable to serious crimes, as if they form part of a continuum:

> "Insecurity today takes various forms in the everyday lives of French people: from uncivil behaviour in public transport to more serious violence against people and trafficking—notably narcotics—at the bottom of buildings, urban violence and brawls between gangs."[14]

This concept of "insecurity" is not neutral in a French context: anyone who lives in France, or knows the cultural environment, will immediately understand that all of the examples above are associated with non-white French people, or those of immigrant origin living in major cities. What is particularly interesting and perhaps unique to France is the way "insecurity"—read: social disorder, violence and crime—coalesces with terrorism as if they are the same thing or have the same foundations. After detailing the increased recruitment of police officers under the Macron government, the law references previous "protective measures" taken by the party to reinforce "internal security" and "fight against terrorism."

Many countries, especially those with a colonial past, characterise non-white minorities—or racialised groups—as the source of crime, so, in a sense, the extract above is to be expected, but the language is important, especially in terms of the word "insecurity." This word makes the dominant perspective, or intended audience, explicit while emphasising the law's emotional foundations. French media reports often mention the words "precarity" and "insecurity" to convey a generalised feeling of instability, or economic fragility in the country. Here, the Macron law makes a connection with this generalised malaise, which predates the law, to imply that it derives from the contemporary behaviour of the law-breaking—and violent—minority, which is racialised as non-white and Muslim (or "Islamist") without providing any evidence as to why this might be the case.

Two brutal terrorist attacks in and around Paris also occurred at the same time as the passage of the "Global Security Law" on 20 October, 2020. Both were linked to *Charlie Hebdo* cartoons. On the September, an 18-year-old man from Pakistan attacked two people with a meat cleaver outside the former Charlie Hebdo headquarters; eight other suspects were also arrested in relation to the attack. The accused left his Punjabi village as an unaccompanied minor in early 2018, travelling alone to Europe, where he allegedly expressed support for the January 2015 *Charlie Hebdo* attacks that killed 12 people and injured 11 more (BBC 2020). Three weeks later, on 16 October, the History-Geography teacher, Samuel Paty, was decapitated by an 18-year-old Chechen man, Abdouallakh Anzorov, who was fatally shot by police at the scene. Paty was murdered outside the middle school, where he, in a class on freedom of the press, had previously shown some of the *Charlie Hebdo* cartoons to students.

My motivation in mentioning the above social context is not to suggest that the people sleeping rough on the streets of Paris are in some way associated with terrorist violence in

---

[13]　Salaire moyen en France (2020). https://fr.jobted.com/salaire#:~:text=Salaire%20moyen%20en%20France%20(2020)&text=Selon%20l\T1 \textquoterightInstitut%20National%20de,300%20%E2%82%AC%20net%20par%20mois). (accessed on 25 November 2020).

[14]　ASSEMBLÉE NATIONALE, PROPOSITION DE LOI *relative à la* sécurité globale, 20 October 2020. Available online: https://www.assemblee-nationale.fr/dyn/15/textes/l15b3452_proposition-loi. (accessed on 24 November 2020).

France. Even if the two most recent attacks were committed by two teenagers who had entered the country through the asylum program, the overwhelming majority of attackers are French born. It does, however, make the longstanding existence of such camps even more surprising. You would think, considering the concern about terrorist violence, that it would be an urgent priority for the French State for this issue to be resolved.

## 2. On the Streets of Paris: The Experience of Displaced Migrants and Refugees

"The police often come in the night, open the tent when I am sleeping and spray tear gas in my tent and face. I wake up and feel like I'm suffocating. It is a feeling of panic."

A Guinean subject quoted in *Still on the Streets*, Refugee Rights Europe. (Stanton 2018)

Between 27 and 30 January 2018, researchers from the London-based NGO, research and advocacy organisation, Refugee Rights Europe, conducted research with migrants and refugees sleeping outside in Paris. They interviewed 283 people—in their native languages—10% of the "estimated 2950 refugees and displaced people thought to be sleeping rough in Paris at the time of the study.(ibid. p. 5)" And yet, as the researchers note in the report, these figures are not certain: "there is uncertainty about the exact population of displaced people in Paris since it is in constant flux."

The mostly adult interview group included 58 children—5.8% of the research sample—and ranged in age from 14 to 52. The majority of those interviewed were men and boys—there was only one woman included—"as those were the demographic groups visible and accessible in the streets of Paris at the time of the study."

"Given the volatility of the situation in Paris" and the fact that it was not a fixed camp, or stable living situation, the researchers tried to interview as many people as possible through a "so-called snowball sampling." Interviews were conducted in the following languages: Amharic, Arabic, Pashto, Persian and Tigrinya. The stated nationalities of the respondents, from the largest number of responses to the least were as follows: Sudanese (35.2%) Afghan (22.1%), Eritrean (10%) and, then in decreasing importance, all numbering less than 10%, there were people from Guinea, Chad, Ethiopia, Somalia and Mali. Other nationalities/countries included in even lower numbers were Yemen, Côte d'Ivoire, Iraq, Libya, Morocco, Senegal and then even smaller numbers from Algeria, Cameroon, Egypt, Nigeria, Pakistan and Sierra Leone. Almost half of the interviewees were aged 18–25 (46%), the second and third largest groups were aged 26–35 (29.7%) and 17 or under (20.5%). More than 85% of the respondents were alone in Paris, and more than 60% (63.6%) had been in Europe for more than six months.

Significantly, a palpable majority of the respondents (more than 85%) had been in another European country before arriving in France. While this is expected in a geographical sense—refugees and migrants from Africa usually enter Europe via Italy—this fact would discount their capacity to claim asylum in France under the Dublin II Convention. Almost 70% of the group had been in Italy, while smaller numbers stated that they had been in Germany, Spain, Sweden and Greece.

The Refugee Rights Europe survey followed a series of set topics relating to feelings of safety and well-being. This included questions relating to police harassment, living conditions, information regarding their time in Europe, and the status of their refugee claim (if applicable). The experience of children, or minors, was covered separately. The negative aspect of this approach—that often relied on set phrases where the interviewees expressed degrees of agreement—is that it avoided any analysis of the reasons for why large numbers of people have ended up in these circumstances in the French capital, or what it might say about weaknesses in the country's assessment of refugee claims, or response to people in an emergency situation.

However, as a record, or snapshot, of the experiences of this group—that are so often not heard—the report is valuable. One theme pre-dominated in the report: harassment by the French police. One-third of the respondents experienced police violence (33.8%), which included a persuasive majority stating that they had been tear-gassed by police

(86%), while another 28% had experienced verbal abuse and just over 20% physical abuse (ibid. p. 11). Some of the children spoke of physical violence and intimidation by the police:

> "One 15-year-old Sudanese minor explained that he was accused of stealing a phone whilst travelling on the underground train. He told researchers he was beaten and taken to the police station where he was left to sleep naked on the floor like an animal." (ibid. p. 13)

Another sixteen-year-old from Guinea recalled:

> "They found me under the La Chapelle bridge, there the police found us lying down and they started spraying tear gas. Then I asked the police where I could sleep, they took my papers and they didn't give them back, they said that they had lost them." (ibid. p. 27)

The researchers were shocked by the living conditions of the migrants and refugees, who had little or no "access to sanitation facilities, (and were) relying on food donation points (provided) by local organisations and civil society." More than 85% of the respondents were sleeping in tents provided by NGOs, "or sleeping under bridges or on damp mattresses on the floor." A few days after the interviews were completed, the "weather deteriorated in Paris and the tents were covered in several inches of snow (ibid. p. 14)."

Health issues were also a concern—a longstanding problem predating the report. Between 2015 and 2017, there were recurring outbreaks of scabies, for instance, which led sanitation workers to refusing to work in the immediate vicinity (C.B. 2017). "It's very difficult to know the exact number of cases," one volunteer working for the charity Emmaüs said in June 2017, but the outreach health service offered by Médecins sans frontières had diagnosed more than 150 cases since the previous December. "To care for people efficiently on the footpaths is a huge challenge, because of the overcrowding and serious issues regarding sanitation (*promiscuité est grande et l'insalubrité importante*)".

More recently, Médecins Sans Frontières has emphasised the risk of contracting COVID-19 for those sleeping on the streets of Paris, including migrants and refugees. MSF has been working with the latter group—offering health care and connecting people with legal advice since 2015. They describe their work on their site:

> "Following the evacuation of Calais, Médecins Sans Frontières turned its attention to Paris, where hundreds of people were arriving each day in the first few months of 2017. Despite the opening of an orientation centre by the Town Hall of Paris in the north of the city, at Porte de la Chapelle, the capacity of the centre was soon overwhelmed. The under resourced help on offer and complexity of the asylum system led to the establishment of multiple camps in the capital, which were regularly dismantled by police without any solution offered in terms of resettlement—an infernal cycle that continues until this day."[15]

In June 2020, the NGO launched a campaign to raise awareness about the treatment of unaccompanied minors seeking asylum in France, especially in terms of the country's ongoing COVID-19 crisis. From March to May 2020, MSF and Médecins du Monde (MdM) completed almost 400 medical consultations and more than 730 psychological sessions, with the organisation COMEDE, for unaccompanied minors with a claim in process.[16] "Throughout, the refusal of government agencies to recognise their status as children and threat of being sent back negatively affected their health," Médecins Sans Frontières stated. Despite multiple messages from both organisations about the sanitary, medical and psychological risks to the minors in Paris, no specific housing arrangements were set up for the group.

---

[15] Médecins Sans Frontières, "France: assistance aux personnes en migration" undated website copy https://www.msf.fr/decouvrir-msf/nos-operations/france-assistance-aux-personnes-en-migration. (Accessed 26 November 2020).

[16] Médecins Sans Frontières "Covid-19: un bilan inacceptable pour les mineurs non-accompagnés en France" 28 May 2020. https://www.msf.fr/communiques-presse/covid-19-un-bilan-inacceptable-pour-les-mineurs-non-accompagnes-en-france.

"So Médecins Sans Frontières funded a shelter in hotels for more than 170 minors in Paris, Bordeaux and Marseille. In Paris, another 107 minors would have spent the lockdown on the street, if not without the help of NGOS (Paris d'Exil, TIMMY—Soutien aux Mineurs Exilés, les Midis du MIE, La Casa et Utopia 56)." Much of the accommodation organised by the French government was inappropriate for the specific needs and vulnerabilities of this group, however.

> "In Paris the sole housing solution offered after six weeks of lockdown was in a gymnasium that was ill-equipped in terms of health and safety recommendations and thought to be a way to reorient the minors towards assistance offered adults in precarious situations. Thus, despite the announcement of the lockdown, there are still hundreds of children on the street in the middle of a health crisis, ignoring all advice regarding the health recommendations."[17]

Of the 58 children interviewed by Refugee Rights Europe in 2018, more than one-third were aged 16 (39.7%), with almost equal numbers aged 17 and 15 (31% and 27.7%, respectively). The group included some 14-year olds. Most of the children came from Guinea (25.9%), with decreasing numbers coming from the following countries: Sudan, Eritrea, Afghanistan, Mali and Chad.

Strikingly high numbers of the group had been in France for more than six months (91.2%), with most of the minors having spent six months in Europe, though some had been in the region for two years. All minors were assessed by the Red Cross, though the researchers noted a "growing concern that the minors were not given the opportunity to have a fully-fledged interview, but are rather assessed within five minutes based on their size and how old they look (Stanton 2018, p. 30)." Refugee Research Europe understands that those found to be minors but aged over 15 were accommodated in hotels, where they were told to leave each day at 7 am and return in the evening.

Few of the minors had applied for asylum in France—only 26.3% of the group (73.7% had not). This might explain why a small majority of the respondents thought that the UK would be the best permanent settlement option for them (at 39.7% of respondents, rather than 37.9% preferring France). Both the low number of asylum applications and uncertain responses regarding settlement indicate a degree of confusion, which might be linked to what appears to be minimal, if any, legal advice and casework support. Of the children surveyed, the vast majority were alone in France (84.5%), though some had family in the UK (66.7%) and other countries in Europe (Switzerland and Germany).

## 3. The Political Context and New Immigration Law

Despite a campaign promise of President Macron that his government would get refugees "off the streets, out of the woods," the expulsion of the migrants and refugees from Place de la République in late November indicates that this issue is not resolved.

One explanation for the apparent stalemate over this issue is "confusion" about which branch of French government is responsible for finding a permanent solution to the problem, with the Town Hall emphasising that the camps reflect core issues within the country's asylum assessment program, and the national government not appearing to engage with the issue at all (outside the tweet from the Interior Minister after the expulsion at the Place de la République in November, 2020, throughout the research for this article, I did not find any earlier statements from the Minister of the Interior—or any other senior politician—regarding the camps in Paris over the past five years).

In contrast, the mayor of Paris, Anne Hidalgo, and Dominique Versini, the city's assistant mayor who oversees "solidarity, the fight against exclusion and refugees", have repeatedly spoken about the issue. In 2019, Versini recalled "a young Somali woman, who

---

[17] A Paris, l'unique solution d'hébergement proposée l'a été après six semaines de confinement, dans un gymnase, inadapté aux consignes sanitaires, et pensé comme un sas pour réorienter ces mineurs vers les dispositifs pour adultes en situation de précarité. Ainsi, malgré les effets d'annonce, ce sont des centaines de mineurs qui sont restés à la rue en pleine crise sanitaire, en dépit des consignes de confinement. https://www.msf.fr/communiques-presse/covid-19-un-bilan-inacceptable-pour-les-mineurs-non-accompagnes-en-france. (accessed on 25 November 2020).



was soon to have a baby, telling her about her life in a camp near la Chapelle, living in terror, pregnant, with her nine-year-old son and husband, who had received status as a political refugee." She affirmed that "as long as refugees continue to gather in the capital, the right to shelter for each person should be unconditional."

Further, Anne Hidalgo, the Socialist mayor of Paris, as journalist Rory Mulholland writes in *The Local*, "(put) the blame for homeless migrants squarely on the government."

> "(Hidalgo) paid a visit to the camps at Porte de la Chapelle and Porte d'Aubervilliers on Tuesday (26/03/19) and said she was shocked at the "migration crisis in the northeast of Paris" where people were "forced to live in inhumane conditions."

> "I do not understand why the state lets indignity and chaos prosper at the gates of the capital of France," she said, vowing to return to the camps every week until the situation was resolved."

> (Mulholland 2019, as cited in [The Local 2019](#))

In 2020, Dominique Versini echoed this idea that the reception and treatment of those sleeping rough on the city's streets was the responsibility of the national, rather than local, government implying that the issue was a direct result of France's asylum regime. "We've asked the government to look at the asylum requests for those who have already failed an asylum request in another E.U. Member State—as it stands, they have to hide for 18 months, sometimes on the street, even if they ultimately have a good chance of obtaining asylum and staying in France ([France 24 English 2020](#))."

In a 2019 report in *Le Parisien,* Versini further explained the measures that the City of Paris had instituted to help the migrants and refugees on the streets of the city. Asked what the Paris government was doing to help the migrants in the north-east of the city, she responded:

> "It's very concerning that the position of the (national) government on this subject has hardened. For the past five years, the City has pushed the State to better manage the migration crisis, but there are political divergences on the matter. These people live in intolerable conditions—a hell—and as long as we welcome them with short-term humanitarian solutions, the State responds that we're offering incentives (*un appel d'air*) . . . As if people would come from Somalia just to get a free meal and use the bathrooms at Porte de la Chapelle! We do what we can to respond to their fundamental, basic needs." ([Beaulieu 2019](#))

She also detailed the assistance that the Town Hall offered the migrants and refugees: 23,000 housing places a year; a humanitarian "welcome system" co-organised with the city of Saint-Denis (Seine-Saint-Denis), while making municipal buildings available, alongside other new propositions which were on "stand-by". The City also helped finance services offered by the NGO France Terre d'Asile that helped the most vulnerable, including a centre on Boulevard Henri-IV in the 4th arrondissement that was entirely financed by the Town Hall.[18]

This position that the camps are solely a consequence of French asylum policy and inaction on the national government is not a view shared by all those familiar with the subject. Utopia 56, for one, stopped working with the Paris Town Hall in 2017 because of the introduction of a policy that meant that help would be conditional on the person accepting to be finger printed—to check against EU records of asylum applications if the person had previously applied in another member state—and therefore potentially be at risk of immediate deportation.[19]

In September 2018, Macron's government passed an asylum/immigration law (*loi pour une immigration maîtrisée, un droit d'asile effectif et une intégration réussie*), which, as with his government's recent laws, provoked an extremely negative reaction. Immigration

---

[18]  "Ces campements de migrants".

[19]  Utopia 56, "Notre Histoire"—Utopia 56 [http://www.utopia56.com/fr/utopia-56/notre-histoire](http://www.utopia56.com/fr/utopia-56/notre-histoire). (Accessed 24 November 2020).

lawyers went on strike for 28 days in opposition. Others, such as Jacques Toubon, France's Défenseur des droits (the independent authority that oversees the country's human rights record), and CIMADE, the country's key NGO safeguarding the rights of asylum seekers and migrants, both condemned the proposed changes.

The law's main features, relevant to this inquiry, were as follows: a reduction in the initial assessment period for asylum claims to six months (in contrast to the previous nine month period) and an increase in the maximum immigration detention limit to 90 days, from a 45 day limit that included families with children, while allowing six-month work rights for people as they wait for the decision on their claim. Applicants now needed to lodge their claim within 90 days, as opposed to an earlier 120-day limit (the length of time available for rejected asylum applicants was also be reduced to two weeks instead of one month). The law removed the possibility of people seeking a reassessment of their claim if they came from designated "safe" third countries; this also applied to people considered to be a serious threat to public order.

Baumard (2018), writing in *Le Monde*, commented how the rationale for the law, as spelled out in its summary, was to "better welcome refugees" and "expel them" (« *mieux accueillir les réfugiés* » *et* « *mieux renvoyer* »). She quoted Serge Slama, professor of law at University Grenoble-Alpes, who stated that the law was, in many respects, tougher than earlier laws on asylum and immigration, proposed by previous Interior Ministers such as Nicolas Sarkozy or Charles Pasqua. "In 2006, it was a matter of privileging "select migration" which included regularising workers without authorisation—*sans papiers*—now it's about toughening procedures, more generally, including the rights of asylum seekers," he said.

It is too early to assess the potential impact of this law, including how it might benefit those currently sleeping on the streets of Paris, or lead to a situation where such people are no longer forced to endure such conditions. However, the argument that this crisis will be resolved when, and if, the asylum process is sped up—the law did not indicate how this will happen, with some condemning the absence of any concrete measures that might improve the system's overall functioning—is unconvincing. The existence of thousands of people sleeping rough on the streets of Paris is as much a housing issue as an immigration problem; moreover, the police violence and intimidation these people face will not end on the law's passage.

Indeed, as some critics of the law have stated, the increased use of detention at the start of the applicant's claim might in fact lead to intensified police harassment, as agents increasingly turn to custodial options as the first recourse. As Médecins Sans Frontières noted in 2019, the "sole promise" that has been kept by authorities was the city's Police Prefecture announcing a policy of "zero return" at Porte de la Chapelle, in the city's north, which is a complete focus on destroying any chance that asylum seekers and migrants might return to the area. "None of this resolves the issue of people on the street," the NGO said, "nor does it allow those who want to make an asylum claim to do so in appropriate conditions (Sans Frontières 2019)."

**Funding:** No external funding.

**Conflicts of Interest:** The author declares no conflict of interest.

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
