# Peer review of "On the Streets of Paris: The Experience of Displaced Migrants and Refugees"

_socsci, doi:10.3390/socsci10040130_

Round 1
Reviewer 1 Report
Dear authors
Thank you very much for producing this manuscript entitled "On the streets of Paris: the experience of displaced migrants and refugees" aiming at both presenting the broader social context and also describing current events in Paris.
First of all, I would like to suggest you check the definition of migrant provided by the UN Migration Agency (IOM), before writing a whole piece of research work. Migrant is any person who is moving or has moved across an international border or within a State away from his/her habitual place of residence, regardless of (1) the person’s legal status; (2) whether the movement is voluntary or involuntary; (3) what the causes for the movement are; or (4) what the length of the stay is.
The whole document lacks a detailed evaluation and analysis. Most of the arguments are unjustified and not supported by evidence.
There is not a distribution by sections offering to the reader an easy structure to follow (for example introduction, methodology, results, discussions and conclusion). Even following a self-structure, the manuscript misses discussions with the collected evidence and main conclusions with the highlights of this work. In terms of structure, it is also missing "bibliography".
The text needs several revisions in editorial terms before being published as research work.
Author Response
The author thanks to reviewers for their review and has revised the manuscript.
Reviewer 2 Report
Article has been clearly cut and pasted directly from a thesis. Unacceptable. Perhaps he/she should seek academic advice about how to properly submit a paper and avoid self plagiarism.
Author Response

(The authors gave the same response as above.)

Reviewer 3 Report
The topic of the 'migration crisis' is still valid and important, so the advantage of the article is that an attempt was made to describe the situation in Paris in this respect.
The authors of the text show examples of brutal police actions against immigrants and refugees. To what extent are they due to the fact that they concern immigrants and refugees, and to what extent is this "normal" behavior of the French police? How is French society responding to such situations? What are the measures taken to prevent the negative effects of the migration crisis? Who and how helps immigrants and refugees in Paris?
First of all, the article requires improvement and developing an appropriate structure. The text lacks an introduction, the part devoted to the research methodology or the presentation of the empirical basis of this study is missing, there is also no research hypotheses, conclusions, summary and recommendations. Currently, it seems to be more journalistic than scientific.
Author Response

(The authors gave the same response as above.)

Reviewer 4 Report
This article makes an important contribution to knowledge in providing a thick description of the insecurity asylum seekers, refugees and migrants are experiencing in Paris and crucially makes the argument that a particularly important feature of the situation in France is the way that insecurity, defined as social order, violence and crime coalesces with terrorism. This allows two key effects to take place. Police responses to the camps become more extreme and new laws are passed with questionable rationale and provisions. This is a key analytical contribution and should be highlighted in the abstract and introduction and in a new conclusion section.
This paper can be strengthened by a tighter focus on key points. To this end the introduction should be tightened and in particular the discussion around visibility/invisibility either removed or, if not removed reintroduced and expanded in the discussion around the way in which notions of insecurity are being used by the government. If this section is kept and expanded a key question for me as a reader is whether the positions of some of the camps - particularly at the Place de la Republique and Marie de Paris are being chosen specifically to raise visibility of their plight - to force their insecurity to be visible to the public and not hidden beneath bridges and atomised. If this is the case then my next question becomes is this a way the migrants, refugees and asylum seekers are trying to have some control over the narrative of what insecurity actually is, when to date this narrative is being used against them?
The flip-flop response of Gerald Darmanin detracts from your main argument. I would strongly recommend it is removed and that section ends on the key point that the clearances are commonplace and longstanding but for the first time police brutality is in the spotlight.
The discussion around the new immigration law's strict rules at the end of the paper can do more to highlight how this also adds to the insecurity and criminalisation and sense of social disorder - helping the reader to make the connections is very important
Some other editing points to do with chronology will help provide greater clarity:
- a clearer timeline of events. I suggest the two paragraphs detailing the protests and response to the global security law be moved to after the long quote from the speech by Macron. Also before introducing the speech provide the context of the terrorist attack a few days before (as opposed to having it after). Also move discussion of the 'reinforcing republican values law' to a footnote so as not to break the chronological flow.
- Both to help with timeline clarity and to maintain key focus I suggest deleting the two paragraphs around Charlie Hebdo reporting. So delete the paragraphs starting "In December, 2020, fourteen" and the paragraph starting "Each day the French newspapers". The next paragraph should then start: In the lead up to the Global Security Law's passage two (nb: note the difference between two and 'a number' which is the current text) terrorist attacks occurred in and around Paris. Both were linked to the Charlie Hebdo cartoons and the trial, then taking place, of the perpetrators of that 2015 attack. On the 25th of September etc...
- Please check the tenses around the September 2018 immigration law you write about it as if it has not been introduced yet with tenses like "it will" but if it was introduced in September 2018 then should this not be "it has"? And early assessment of its results is perhaps possible? Also in this section the English translation the Baumard quote to "re-send them" needs greater context - I am not clear what that means.
In footnote 20 you have a note to the editors which should be removed. The rationale for your argument is clear however, as you suggest, references at this point providing evidence (in a footnote) would be excellent. Particularly the statement "generally seen to be" should only be stated when referenced with evidence.
Finally two typos. On page 13 of the pdf, in the second last paragraph starting "Strikingly" you state minors aged over 15, is this meant to be under 15? In your final sentence you have written NOG instead of NGO.
I look forward to seeing the final version of this article.
Author Response
Reader’s report 4
- Ending of section on Darmanin quotes: I have left it because I think that it works quite well to have this sense of the cynical, even opportunistic nature of French politicians in terms of this issue. If it’s still felt that it doesn’t work as it is, they could be cut, or turned into a footnote.
- I spent some time changing the order of the text regarding the French laws/discourse, I am not sure that it is clearer now and wonder if it might be better to cut the entire section (I have included a second version of the article without that section). I am happy to hear your thoughts as to whether this might be better. I am fine with either option.
- I have also added material to further develop the significance of the initial protest and also elsewhere relating to the multiculturalism footnote.
Reviewer 5 Report
Review of On the streets of Paris: the experience of displaced migrants and refugees
I think this is a good paper that discusses the treatment of displaced persons on the streets of Paris against a backdrop of the contemporary social and political context. It is timely, nicely constructed, well-written and well-connected to highly-relevant themes of insecurity and French identity. I would encourage its publication subject to the authors’ responses to the following suggestions, provocations, etc.
- Pages 4-5 fn 9 - I beg to differ with the authors’ translation of communautarisme into English as akin to multiculturalism. In French, and for republican French people, communautarisme (better translated as “communalism” to my mind) is a loaded term and tends to infer the splintering or fracturing of French society into communal groups based on religious, cultural, or ethnic affiliations.
- Such communal (communautaire) attachments are, in many Republican camps, considered a major threat to the French republican common project of living together, le vivre ensemble, and such cultural minorities are frequently stigmatised. Cf Matteo Gianni, 2013 "The Democratic Integration of Difference: Reflections on the Paradoxes of the French Republican Model of Citizenship." In Spheres of Global Justice, Jean-Christophe Merle (ed.), 391-402 (especially p398).
- This point is significant, since it goes some way to explaining why communautarisme is what the French Republic fears most. And to why, in relation to the authors’ query (p.8), in their comments, ‘President Macron and his government … merge relatively anodyne issues (from my non-French perspective) with the illegal and extremely violent’.
- Page 7 – the para beginning @197 needs to be more evidence-based. In particular, on the implied French president’s position that there is a ‘unified push among “Islamist separatists” - or those with “radical Islamist” views - to either contravene the country’s laws or overthrow them in order to take over the reigns [sic] of power’.
- The discussion on French “insecurity” as a source of generalised malaise and its powerful and ongoing linkage in public debate to questions of asylum and immigration is interesting and welcome here. I think the article would be improved by highlighting that this is far from a new phenomenon in the French context. Moreover, it would be helpful to hear the authors’ thoughts on whether the new “Global Security Law” becomes the French state’s justifiable response to its sustained politics of insecurity? For a concise discussion on contemporary French politics of insecurity, see Taylor, A. (2018). “Crimes of solidarity”: France’s contemporary crisis of hospitality. In A. Maazaoui (Ed.), Making strangers: outsiders, aliens and foreigners (pp. 39–53).(especially @ pp 49-50).
- There is a growing literature on the politics of visibility in relation to asylum seeker communities, especially those sleeping rough on city streets. This might offer fruitful explorations on why “the longstanding existence of such camps” is not that “surprising” and why a resolution of the issue is slow to materialise.
- See, eg: Tazzioli, Martina, and William Walters. 2016. "The Sight of Migration: Governmentality, Visibility and Europe’s Contested Borders." Global Society 30 (3):445-64.
- I think one last para at the close of the article that more forcefully state the authors’ position and conclusion is needed.
Typos & translations:
p1 @33: Gendarmerie nationale
p3 @71: “(of such…)
p12 @374 & elsewhere: City of Paris is the better translation of la mairie de Paris (unless when referring to the actual edifice of the City Hall, in which case: Paris City Hall)
p12 @ 363,367 & 386 (&529): Capitalise ‘Médecins sans frontières’
p15 @483: arrondissement not italicised (where it is, p2)
p16 @ 510: [les] renvoyer is better translated as “send them back” or “expel them”
p17 @534: NGO
Author Response
Reader’s report 5
- I have integrated many of your comments re word usage/translation into the text; thank you too for your very interesting comments on multiculturalism and how best to translate the word and concept This is complex and worth much more than a footnote or passing reference. I have to admit that my experience as a non-French person who has lived in France for a long time makes it difficult for me to understand the urgent concern so many people in France feel about “secularism” (I have an informal understanding, based on conversations with French people, who place it in a historical context). Often it seems to me as an excuse for xenophobia or political opportunism. I see parallels with far-right populism in countries such as Australia, as mentioned in the amended footnote but then many of the strongest supporters of the laws, and arguments in France are on the Left. I understand your idea of using the word “communalism” but wonder if this would be clear to English speakers; the word multiculturalism as I mention is also used as a slur among certain people, or political parties.
- Thank you for your academic references, unfortunately I don’t have access to a database to find these articles, but I will definitely seek them out for later pieces of writing.